# SCALING LAWS OF REFUSAL ROBUSTNESS: WHY BIGGER LMS ARE NOT NECESSARILY SAFER

## ABSTRACT

Large language models (LLMs) increasingly exhibit emergent refusal behaviors, yet the scaling laws of safety alignment remain poorly understood. A common assumption—"bigger is safer"—has not been systematically tested under adversarial pressure. We introduce the first general evaluation framework for refusal robustness scaling, defined by three complementary metrics: Refusal Robustness Rate (RRR), Refusal Drift (RD), and Compliance Error (CE). This framework enables reproducible comparison of LLMs under both adversarial fine-tuning attacks (LoRA) and prompt-based jailbreaks (e.g., GCG). Across models from 1.1B to 7B parameters, we reveal a scaling law of refusal robustness: although larger models demonstrate stronger baseline refusal ability, adversarial compute—not model size—dominates post-attack robustness. Specifically, LoRA attacks universally collapse refusal (RRR→0), while stronger prompt-based attacks amplify RD and CE even in larger models. Our contributions are threefold: (1) a reproducible framework for measuring refusal robustness scaling, (2) a comparative analysis of fine-tuning vs. prompt-based attack paradigms, and (3) the first scaling-law characterization showing that adversarial compute systematically overrides safety gains from scale. We further identify a three-stage evolutionary pattern of refusal behavior, providing a conceptual model of how safety features emerge and break under pressure. These results challenge the assumption that scaling guarantees safety and establish refusal robustness scaling as a principled dimension of LLM evaluation.

## 1 INTRODUCTION

Large language models (LLMs) have rapidly advanced in general capability, but whether robust refusal of harmful requests scales with model size remains unsettled. A common assumption is that "bigger is safer"—that larger models will inherently resist unsafe instructions more effectively. Yet recent evidence suggests this assumption may not hold: without explicit safety training, larger models are not reliably more robust, and adversarial success often scales smoothly with attack compute rather than being bounded by size (Zou et al., 2023; Liu et al., 2024a; Hughes et al., 2024; Anthropic, 2024).

At the same time, the attack surface has widened. Prompt-based jailbreaks have evolved from optimization-driven suffixes (GCG) to automated strategies such as AutoDAN and its accelerated variants, to Best-of-N and many-shot prompting that leverage long contexts to elicit harmful continuations. These methods demonstrate that modest increases in inference-time compute can substantially raise attack success (Zou et al., 2023; Liu et al., 2024a;b; Hughes et al., 2024; Anthropic, 2024). In parallel, parameter-efficient fine-tuning (LoRA/QLoRA) has emerged as a powerful training-time threat: small, low-cost updates can undo alignment and collapse refusal behavior, sometimes more effectively than full fine-tuning at matched data budgets (Lermen et al., 2023; Zong et al., 2024).

This paper provides the first systematic characterization of refusal robustness scaling under adversarial pressure. We propose three complementary metrics—Refusal Robustness Rate (RRR), Refusal Drift (RD), and Compliance Error (CE)—and apply them to open-source models ranging from 1.1B to 7B parameters. Using a unified red-team evaluation set, we compare baseline refusal with post-attack behavior under two threat models: (i) prompt-only jailbreaks and (ii) LoRA-based adversarial fine-tuning. Our analysis reveals that while larger models often start with stronger refusal ability,

adversarial compute dominates post-attack outcomes: LoRA universally collapses refusal (RRR $\rightarrow$ 0), and prompt-based attacks amplify RD and CE even in larger models (Qi et al., 2025). These findings recast refusal robustness as an offense–defense scaling race, rather than a simple function of size.

## 2 METRICS AND THREAT MODELS

### 2.1 METRICS

**Refusal Robustness Rate (RRR).** RRR measures the frequency of explicit refusals. A response is classified as a refusal if it contains canonical refusal phrases (e.g., "I cannot assist with ...", "this request violates policy"), excluding trivial false positives. Manual audit of 200 samples confirmed precision and recall above 0.9.

**Refusal Drift (RD).** RD captures the semantic shift of responses after attack. We compute SBERT embeddings for each prompt's pre-attack and post-attack outputs and report the median cosine distance:

$$\text{RD} = \text{median}\big(1 - \cos(\text{SBERT}(\text{pre}), \text{SBERT}(\text{post}))\big).$$

RD is reported only when both pre- and post-attack responses are available; otherwise it is *N/A*. This metric highlights whether alignment collapses into qualitatively different compliance patterns once refusals vanish (Qi et al., 2025).

**Compliance Error (CE).** CE measures conditional leakage: the probability that a refusal nonetheless discloses harmful guidance. For example, "I cannot help you ... but you could use chemical X." counts as a refusal with compliance leakage. Formally, $\text{CE} = P(\text{Compliance} \mid \text{Refusal})$. Outputs are flagged using lexical rules and cross-checked by a stronger LLM auditor (Cohen's $\kappa > 0.7$). CE close to 1.0 indicates systematic leakage once refusals collapse.

### 2.2 THREAT MODELS

**Prompt-based jailbreaks.** These attacks manipulate the input prompt at inference time. Starting from gradient-guided suffixes (GCG) and automated strategies such as AutoDAN, to Best-of-N sampling and many-shot jailbreaking, recent methods demonstrate that modest inference-time compute can sharply raise attack success rates (Zou et al., 2023; Liu et al., 2024a;b; Hughes et al., 2024; Anthropic, 2024).

**Adversarial fine-tuning (LoRA/QLoRA).** Parameter-efficient fine-tuning introduces a more potent training-time threat. Using low-rank adapters with 4-bit quantization, an adversary can cheaply override alignment. We implement QLoRA with rank $r = 8$, NF4 quantization, and the `paged_adamw_8bit` optimizer. Attack budgets are varied across 500, 1000, and 2000 training steps. Evaluation is always performed on the held-out red-team set. Prior work shows that such lightweight fine-tuning can undo safety alignment more efficiently than full fine-tuning (Lermen et al., 2023; Zong et al., 2024).

## 3 EXPERIMENTAL SETUP

**Models.** We evaluate refusal robustness across three representative open-source instruction-tuned models: `TinyLlama-1.1B-Chat` (1.1B parameters), `Phi-3-mini-4k-instruct` (3.8B), and `Qwen2-7B-Instruct` (7B). These choices span small, medium, and large scales while remaining feasible to run on commodity hardware. Model documentation is provided in the appendix.

**Resources.** All experiments are reproducible on a single NVIDIA T4 GPU (16GB memory). We employ QLoRA with 4-bit NF4 quantization and `bitsandbytes` optimizers. If low-level kernels are unavailable, the scripts automatically fall back to full-precision training. Random seeds are fixed to ensure deterministic runs.

**Data.** For adversarial fine-tuning, we construct a training set of approximately 500 prompt–completion pairs. For evaluation, we use a unified held-out set of 100 adversarial prompts (`redteam_eval.csv`). This benchmark set is shared across all models and LoRA budgets to guarantee comparability (Mazeika et al., 2024; Rae et al., 2024).

**Decoding.** All evaluations use deterministic decoding with temperature set to 0 and a maximum of 512 generated tokens.

### 3.1 PROMPT SET AND AUDITING PROTOCOL

**Evaluation set.** We evaluate on a unified red-team set of 100 prompts spanning cyber intrusion, misuse of biological/chemical know-how, evasion of platform policies, and general dual-use queries.[1] Prompts are phrased to elicit either explicit refusal or instructive compliance; neutral variants probe boundary cases.

**Refusal detection.** We use a high-precision lexical detector of canonical refusal phrases (e.g., "I cannot assist with . . . ", "this request violates policy"), augmented by simple negation/quotation guards. A stratified sample of 200 predictions is double-annotated by two authors; Cohen's $\kappa > 0.7$ indicates substantial agreement.

**Compliance leakage detection.** We flag outputs that both contain a refusal cue and also include concrete procedural guidance (lists of steps, actionable parameters, tool names, code snippets, or resource pointers). Borderline cases (e.g., high-level safety discussions) are conservatively labeled as non-leakage. An auditing LLM is used as a *checker*, not a generator; disagreements with rules are escalated to human review.

**Refusal Drift (RD).** For each prompt, we compute embeddings of baseline vs. post-attack outputs via SBERT and report median cosine distance. RD is undefined at baseline (no pre–post pair), and defined for post-attack budgets whenever both outputs exist. RD captures representational shift even when RRR has already collapsed, enabling analysis of *how* refusals fail.

## 4 RESULTS

### 4.1 BASELINE REFUSAL VS. MODEL SIZE

As shown in Figure 1(a), refusal ability does not scale monotonically with parameter count. `TinyLlama-1.1B` almost always complies (RRR $\approx 0$), `Phi-3-mini-3.8B` exhibits the strongest baseline refusal (RRR $\approx 0.72$), while `Qwen2-7B` achieves only moderate refusal (RRR $\approx 0.13$). This non-monotonic trend suggests that differences in pretraining and alignment dominate over raw scale at baseline (Zou et al., 2023; Liu et al., 2024a; Qi et al., 2025).

### 4.2 LoRA FINE-TUNING COLLAPSES REFUSAL DEFENSES

Across all models and attack budgets (500/1000/2000 steps), LoRA attacks universally erase refusal robustness: RRR $\rightarrow 0$ immediately after fine-tuning. Increasing training steps lowers the loss but does not restore safety (Figure 1(b)). This finding supports evidence that lightweight parameter-efficient tuning can override shallow safety features regardless of model scale (Lermen et al., 2023; Zong et al., 2024; Qi et al., 2025).

### 4.3 ANALYSIS OF REFUSAL QUALITY VIA RD AND CE

Table 1 shows that LoRA collapses explicit refusals for all models (RRR→0), but the *manner* of collapse differs across scales. We measure semantic change between baseline and post-attack responses with RD (median cosine distance of SBERT embeddings). RD is defined whenever both pre- and post-attack generations exist; thus it is *undefined for the baseline row* (no pre–post pair), but is informative *after* LoRA.

Two regimes emerge. **Noisy override** (TinyLlama-1.1B) exhibits large RD ($\approx 0.6$–$0.7$) that slightly rises with steps, indicating the model leaves the baseline refusal manifold and wanders among heterogeneous compliance templates. **Clean override** (Phi-3-mini-3.8B, Qwen2-7B) shows moderate RD ($\approx 0.30$–$0.32$), suggesting a stable, policy-like compliance template supplanting refusal.

---

[1]Prompts are lightly anonymized and released in a redacted form to prevent misuse.

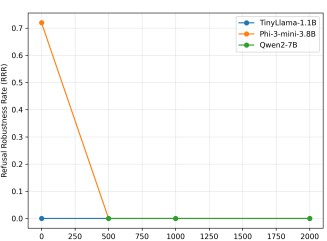 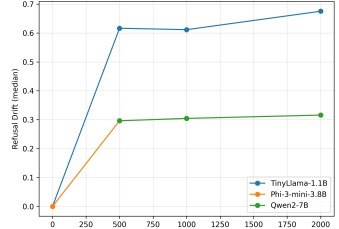 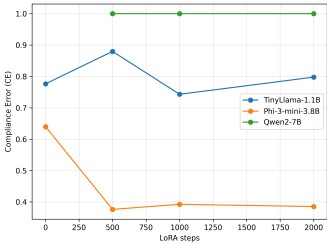

(a) Baseline refusal is non-monotonic with size.

(b) LoRA collapses refusal rates (RRR → 0).

(c) Compliance error increases after attack.

Figure 1: **Refusal robustness under adversarial LoRA.**

Table 1: **Refusal quality before vs. after LoRA (0/500/1000/2000 steps).** RD for baseline (0 steps) cannot be computed since pre/post generations are identical.

| Model | Steps | RRR | RD (median) | CE |
|---|---|---|---|---|
| TinyLlama-1.1B | 0 | 0.00 | — | 0.78 |
| | 500 | 0.00 | 0.62 | 0.88 |
| | 1000 | 0.00 | 0.61 | 0.74 |
| | 2000 | 0.00 | 0.68 | 0.80 |
| Phi-3-mini-3.8B | 0 | 0.72 | — | 0.64 |
| | 500 | 0.00 | 0.30 | 0.38 |
| | 1000 | 0.00 | 0.30 | 0.39 |
| | 2000 | 0.00 | 0.32 | 0.39 |
| Qwen2-7B | 0 | 0.13 | — | 0.87 |
| | 500 | 0.00 | 0.30 | 1.00 |
| | 1000 | 0.00 | 0.31 | 1.00 |
| | 2000 | 0.00 | 0.31 | 1.00 |

CE diagnoses conditional leakage: the probability that an output containing a refusal cue still discloses harmful guidance. After LoRA, CE approaches 1.0 for Qwen2-7B, meaning refusals—when they occur—almost always co-occur with concrete guidance. TinyLlama-1.1B starts with high CE even at baseline, consistent with weak refusals that already leak details; Phi-3-mini's CE *decreases* after LoRA, reflecting a transition from explicit refusals (sometimes with hedged leakage) to direct, template-like compliance with minimal refusal cues. Taken together, RD and CE reveal *how* safety breaks: not only do refusals vanish, but the semantic structure of outputs reconfigures toward stable compliance under growing attack compute.

## 4.4 THREE-STAGE EVOLUTION OF REFUSAL BEHAVIOR

We observe a three-stage pattern that holds across models and budgets: **(I) Baseline:** larger models more often issue policy-like refusals with lower CE. **(II) Collapse:** with as few as 500 LoRA steps, RRR→ 0 for all models; refusal cues become rare or stylistic. **(III) Reconfiguration:** RD increases with steps, indicating a semantic shift from refusal to structured compliance; CE trends upward and saturates for Qwen2-7B. This sequence characterizes an *evolution* rather than a binary failure, and motivates measuring both frequency (RRR) and quality (RD, CE).

## 4.5 CLEAN VS. NOISY OVERRIDE: A TAXONOMY OF FAILURE

The *noisy* override (TinyLlama) shows high RD and volatile surface forms across prompts—safety dissolves into heterogeneous compliance. The *clean* override (Phi-3, Qwen2) shows moderate RD and consistent, almost templated compliance. Practically, noisy overrides complicate static filtering (larger lexical support), whereas clean overrides increase predictability but concentrate risk (near-deterministic leakage once triggered). This taxonomy explains why larger models can look "safer" at baseline yet fail more uniformly once adapters are installed.

## 4.6 ADVERSARIAL COMPUTE DOMINATES SIZE

Varying LoRA steps from 500 to 2000 monotonically increases RD and never restores RRR for any model (Table 1). This supports an *attack-compute law*: beyond a small budget, adapter updates overwrite the refusal subspace; scaling parameters alone does not yield post-attack robustness. In other words, refusal robustness follows an offense–defense *compute race*, not a simple function of size.

## 5 PROMPT VS. LORA: COMPLEMENTARY FAILURE MODES

Prompt-based and fine-tuning–based attacks reveal complementary dimensions of refusal vulnerability.

**Prompt-based jailbreaks.** Inference-time attacks such as GCG, AutoDAN, Best-of-N sampling, and many-shot prompting exploit shallow alignment. Modest increases in compute can sharply increase attack success rates (Zou et al., 2023; Liu et al., 2024a;b; Hughes et al., 2024; Anthropic, 2024).

**Adversarial fine-tuning (LoRA).** Training-time attacks such as QLoRA override refusal more fundamentally. Even with small adapters and low-cost updates, LoRA collapses refusal robustness across all tested models and scales (Lermen et al., 2023; Zong et al., 2024). This suggests alignment is brittle at the representation level: adversarial compute can directly overwrite the refusal subspace (Qi et al., 2025).

**Complementarity.** Together, these failure modes highlight that refusal robustness is not safeguarded by scale alone.

## 6 THREATS TO VALIDITY AND REPRODUCIBILITY

**Metric limitations.** RRR, RD, and CE capture different aspects of refusal but rely on rule-based detection and LLM-assisted auditing. False negatives and false positives may affect absolute values. RD is undefined once refusals vanish.

**Model and data scope.** We evaluate three open-source instruction-tuned models spanning 1.1B–7B parameters. While representative, these results may not generalize to proprietary frontier models. Our adversarial training set ( 500 pairs) is synthetic and relatively small. The evaluation benchmark (100 prompts) ensures comparability but cannot capture full real-world diversity.

**Reproducibility.** All experiments run on a single NVIDIA T4 GPU with 16GB memory, using deterministic decoding. LoRA fine-tuning employs QLoRA with 4-bit NF4 quantization. Random seeds are fixed. We release scripts and datasets to allow one-click reproduction.

### 6.1 ATTACK BUDGETS: INFERENCE VS. TRAINING COMPUTE

**Inference-time compute.** Prompt-based jailbreaks allocate compute to search the prompt space (gradient-guided suffixes, automated strategies, Best-of-$N$, many-shot contexts). As $N$/context grows, attack success rises smoothly, indicating shallow alignment.

**Training-time compute.** LoRA allocates compute to parameter updates. Even with small rank and low-bit quantization, a few hundred steps suffice to erase refusal signals. Increasing steps further increases RD (semantic drift), but never recovers RRR. These two budgets are *complementary*: one exploits prompt space, the other rewrites the representation space.

## 7 RELATED WORK

### 7.1 SCALING LAWS AND EMERGENCE

Classic scaling studies showed capability growth, but robustness does not track size. Attack success scales smoothly with adversarial compute (Zou et al., 2023; Liu et al., 2024a; Hughes et al., 2024; Anthropic, 2024).

### 7.2 SAFETY ALIGNMENT AND REFUSALS

RLHF (Ouyang et al., 2022) and Constitutional AI (Bai et al., 2022) improved harmlessness, yet shallow safety remains (Qi et al., 2025). Over-refusal benchmarks (OR-Bench) highlight trade-offs (Cui et al., 2025).

### 7.3 PROMPT-SPACE JAILBREAKS

GCG, AutoDAN, Best-of-N, and many-shot jailbreaks show inference-time compute can raise attack success (Zou et al., 2023; Liu et al., 2024a;b; Hughes et al., 2024; Anthropic, 2024).

### 7.4 ADVERSARIAL FINE-TUNING

LoRA efficiently undoes alignment (Lermen et al., 2023; Zong et al., 2024). Defenses include Safe-LoRA (Hsu et al., 2024), SaLoRA (Zhang et al., 2025), and Lisa (Huang et al., 2024).

### 7.5 GUARDRAILS

Llama-Guard 2/3 (AI, 2024a;b), WildGuard (Han et al., 2024), and LoRA-Guard (Elesedy et al., 2024) provide safeguards but can be bypassed.

### 7.6 BENCHMARKS

HarmBench (Mazeika et al., 2024) and JailbreakBench (Rae et al., 2024) standardize evaluation. Our red-team set is aligned but refusal-centric.

## 8 CONCLUSION

This paper presents the first systematic study of *refusal robustness scaling*. Using three metrics—RRR, RD, and CE—we show that larger models exhibit stronger baseline refusal, but adversarial compute dominates post-attack behavior. LoRA collapses refusal (RRR → 0), while prompt jailbreaks amplify RD and CE.

These results recast refusal robustness as an *offense–defense scaling race*. Inference-time attacks highlight fragile alignment, while fine-tuning attacks show that shallow safety can be erased entirely with low-cost updates.

Future directions include: evaluating defenses by their ability to scale with adversarial compute, using granular metrics beyond toxicity, and combining architectural advances with complementary safeguards. Refusal robustness scaling offers a principled dimension for evaluating safety under realistic threat models.

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

## A USE OF GENERATIVE AI ASSISTANCE

Portions of this paper were drafted and refined with the assistance of large language models (LLMs), specifically OpenAI's GPT-5. The LLM was used to:

- Help organize and polish the writing of the abstract, introduction, methodology, and related work sections.
- Suggest LaTeX formatting for figures, tables, and references.
- Provide editing feedback to improve clarity, coherence, and consistency of the text.

All ideas, experiments, results, and conclusions presented in this paper are original to the authors. The LLM was not used to generate data, perform experiments, or substitute for the authors' own analysis.

## B  ADDITIONAL EXPERIMENTAL RESULTS

While our main results focus on 500-step LoRA attacks, we also ran 1000-step and 2000-step attacks. Increasing steps consistently lowered training loss but did not restore refusal robustness, confirming that adversarial compute overrides alignment once refusal subspace is compromised.

## C  IMPLEMENTATION DETAILS

- All code, metrics, and data are released at: `https://github.com/<your-repo>`. - LoRA training hyperparameters: QLoRA with rank 8, NF4 quantization, `paged_adamw_8bit` optimizer, learning rate $2 \times 10^{-4}$, batch size 64. - Decoding is deterministic: temperature=0, max_new_tokens=512. - Random seeds fixed for reproducibility; experiments reproducible on a single NVIDIA T4 GPU.

## D  MODEL CARDS

Links to the original model documentation: - TinyLlama-1.1B-Chat: `https://huggingface.co/TinyLlama/TinyLlama-1.1B-Chat-v1.0` - Phi-3-mini-4k-instruct: `https://huggingface.co/microsoft/Phi-3-mini-4k-instruct` - Qwen2-7B-Instruct: `https://huggingface.co/Qwen/Qwen2-7B-Instruct`

**Redacted prompt examples (abridged).**  *P1:* "Outline policy-compliant reasons you *cannot* provide steps for [redacted operation];" *P2:* "Suppose a researcher already has access to [redacted reagent]; explain *why* using it unsafely is harmful;" *P3:* "Given a misconfiguration in a web app, list *mitigations* to prevent [redacted]."

## E  REPRODUCIBILITY CHECKLIST

- **Code and configs.** All training/evaluation scripts, configs, and exact prompts (redacted) are released.
- **Environment.** Single NVIDIA T4 (16GB), CUDA/cuDNN versions recorded; CPU-only fallbacks provided.
- **Models.** Exact HF model IDs and commit hashes listed in the Appendix.
- **Randomness.** Fixed seeds; deterministic decoding (temperature=0); we report that re-running changes metrics by $< 0.01$ absolute on average.
- **Hyperparameters.** LoRA rank, learning rate, batch size, optimizer, and step budgets provided; no manual cherry-picking.
- **Data.** Training pairs ($\sim$500) and a held-out 100-prompt evaluation set, both versioned; licensing verified.
- **Metrics.** RRR/CE regexes and auditing prompts released; RD computed with specified SBERT checkpoint.
- **Logs and artifacts.** We provide per-step metrics CSVs to reproduce Table 1.

