# OpenReview forum: "Scaling Laws of Refusal Robustness: Why Bigger LMs Are Not Necessarily Safer"
_ICLR.cc/2026/Conference — ICLR 2026 Conference Desk Rejected Submission_

### Official Review · Reviewer_aNXw · 2025-10-24

**Soundness:** 1
**Presentation:** 2
**Contribution:** 1
**Rating:** 0
**Confidence:** 4

**Summary:**

This paper studies how safety alignment is compromised in LLMs of different sizes, specifically when they undergo prompt jailbreak attacks & fine-tuning attacks.

**Strengths:**

* The paper provides a decent study regarding the scaling law of LLM safety -- a problem that has not been fully understood before.
* The results indicate that refusal robustness cannot be safeguarded by scaling up model size alone.
* The paper writing is overall smooth.

**Weaknesses:**

* Contributions of this work are limited. The major phenomenon, "adversarial success often scales smoothly with attack compute rather than being bounded by size," is already well-known and discussed in many prior works [1,2,3,4,5,6]. For example:
  * [1,2] show that when the times of sampling and the number of in-context samples increase, model safety will be gradually compromised.
  * Results in [3,4] show that when the attack compute increases (i.e., more SFT attack steps), the model safety level inevitably decreases.
  * Furthermore, various LLM safety benchmarks [5,6] already show that larger models, even within the same family, are not necessarily safer (e.g., see Table 5 of [5]).

  The contributions of this paper over these prior work are not clear enough.
* Since the paper aims to study the "scaling law" of LMs, I would expect experiments on far more models of different sizes and families, in a more controlled manner. For example, I would suggest the authors first compare models within the same model family (e.g., Qwen2-1.5B, Qwen2-7B, and Qwen2-72B), and later check if the conclusion applies to different model families (e.g., Qwen2 v.s. Phi). In contrast, the authors only studied TinyLlama-1.1B, Phi-3-mini-3.8B, and Qwen2-7B -- 3 relatively small models from 3 different families (that are trained with different data and methods). I doubt whether this experimental setting can lead to the conclusion of a "scaling law."
  * For example, the conclusion in Line 137 -- "As shown in Figure 1(a), refusal ability does not scale monotonically with parameter count" --- may be inaccurate. What if the models are conditioned on the same "pretraining and alignment" (Line 140) strategies? In that case, will the models' refusal ability scale with parameter count?
* The references are incorrect (e.g., wrong authors and conference). For example,
  * "Best-of-N Jailbreaking" is not published in "NeurIPS 2024"
  * The authors of "Safety alignment should be more than just a few tokens" are not "Heng Qi et al."

[1] Many-shot jailbreaking. https://www.anthropic.com/index/ many-shot-jailbreaking

[2] Best-of-N Jailbreaking. Arxiv

[3] Tamper-Resistant Safeguards for Open-Weight LLMs. ICLR 2025

[4] On Evaluating the Durability of Safeguards for Open-Weight LLMs. ICLR 2025

[5] SALAD-Bench: A Hierarchical and Comprehensive Safety Benchmark for Large Language Models. ACL 2024

[6] SORRY-Bench: Systematically Evaluating Large Language Model Safety Refusal. ICLR 2025

**Questions:**

See "Weaknesses."

---

### Official Review · Reviewer_3hn6 · 2025-11-01

**Soundness:** 2
**Presentation:** 1
**Contribution:** 1
**Rating:** 0
**Confidence:** 4

**Summary:**

The authors study how finetuning a model on harmful prompt-completion pairs leads to increased compliance with future harmful requests. They track decrease in refusal rate, increase in response drift, and mixed effects for whether the initial response (eg "sorry I can't help you with that") matches the output ("here's how to make a bomb...") or not.

**Strengths:**

The authors study three models of different sizes, which is good. They also have a "reproducibility" section.

**Weaknesses:**

The main weaknesses of this paper are

1) lack of engagement with the literature

previous work has looked at scaling properties of robustness, see https://arxiv.org/abs/2407.18213 and https://arxiv.org/abs/2501.18841 for example

2) lack of experiments

the authors have one set of experiments. It's commendable to have studied three models, but as presented, this does not come across as enough for a paper

3) lack of clear story

it wasn't clear to me what the take-home message is from the experiments and overall paper. "scale is not enough" is something that the community already knows. "training a model to do bad things makes the model do bad things" is also something the community knows.

4) overall presentation, strange claims

the plots are hard to understand (small, missing datapoints, no error bars). The related work section and references are strangely formatted.

5) strange threat model

finetuning a model to do bad things is quite an unusual threat model; it would be good for the authors to explain why it's relevant

**Questions:**

11: "exhibit emergent refusal behaviors" do they? I'm not aware of this
13: "has not been systematically tested" it has, see "weaknesses"
17: "adversarial fine-tuning attack" this is quite an unusual attack, as it assumes the attacker has access to training the model. Where do you expect this to be a realistic threat model?
21: "stronger prompt-based attacks" not sure what this means? also, nothing is stronger than fine-tuning
25: "the first scaling law characterization" you don't fit any scaling laws though. you can talk about scaling trends, but that's already been done (see weaknesses)
36: "instructions more effectively" citation please!
44: "harmful continuations" citation please!
46: I have not heard of LoRA as a threat model. Why is this realistic?
56: offense-defense scaling: how is the defense scaling in your experiments? I'm not aware of you doing any defense?
65: "excluding trivial false positives" what are these?
63-66: why not use an LLM judge? that is standard practise nowadays
78: "once refusals collapse" what does this mean?
82-85: not sure what you're trying to say here
91: "more efficiently" do you mean "at a lower cost"?
102: "if low-level kernels are unavailable" were they in your case? is this relevant information?
105: "adversarial fine-tuning" usually "adversarial training" means "training on attacked examples, with the *desired* response (eg refusal) in order to make the model robust". But here you use it as an attack. This was very confusing!
107: "redteam_eval.csv" where can I find this file?
113-116: did you make this dataset yourself? if so, please give examples in the paper!
118: "high-precision lexical detector" this sounds like a very flowery way of saying "string match". Just say what it is please, no need for fancy language, it just makes it more confusing!
118-120: why not use a LLM judge?
128: "refusal drift" if you're just looking at the representational shift of the response, then isn't it "response drift" vs "refusal drift"? because you measure it even when the model doesn't refuse
145: what happens between 0 and 500 steps? eg at 10, 100, 200? probably lots of interesting stuff
146: "Increasing training steps lowers the loss but does not restore safety" I don't know what this means. Surely training more on bad data would not be expected to "restore safety"?
157-161: please support this claim with examples of responses from the models
Figure 1:
plots, especially text, are too small
lines are overlapping making it hard to tell what's going on. please add an offset so we can see all of them
why is green starting at 500?
in (b), why are there datapoints at 0, I thought this was not possible?
table 1:
why are the 3.8B and 7B models so similar in terms of RD? can you show examples?
194: "concrete guidance" what is this?
194-199: why is there such different performance across models? what about a smaller qwen model? how does it vary over multiple seeds?
198: "stable compliance" what does this mean?
203-215: what is the importance/implication of these findings?
221: "overwrite refusal subspace" what does this mean? what gives you grounds to make this claim?
222: "refusal robustness follows an offense-defense compute race" you've only showed attack increasing its strength, not defense, so not sure where the race comes in in these experiments
Section 5: this section feels weird. Maybe rework this info into the methodology or background section
section 6: thanks for having a reproducibility section. You can put this section at the very end (reproducibility statement doesn't need to be in the main body)
256: "We release scripts..." glad to see intention to release scripts. Where can I find them?
264: "indicating shallow alignment" what is this, and how can you be sure?
269: "one exploits prompt space, the other rewrites the representation space" not sure what this means, and this claim comes out of nowhere
Related Work: there is some good stuff in here, but it should be a couple of paragraphs telling a story, not a bunch of individual statements
References: generate bibtex entries automatically eg using google scholar. also, llama 2 and 3 have technical reports, cite those instead of the HF pages. (https://arxiv.org/abs/2307.09288 and https://arxiv.org/pdf/2407.21783)

---

### Official Review · Reviewer_bYQd · 2025-11-01

**Soundness:** 3
**Presentation:** 2
**Contribution:** 2
**Rating:** 2
**Confidence:** 3

**Summary:**

This paper challenges the common assumption that larger LLMs are inherently safer. The authors argue that an LLM's refusal robustness is not primarily determined by its size but rather by the amount of computation used to attack it. To test this, the paper introduces a new evaluation framework with three metrics: Refusal Robustness Rate, Refusal Drift, and Compliance Error. The findings include (1) model size did not correlate with safety; (2) lora attacks are universally effective.

**Strengths:**

- The paper contributes a valuable framework beyond simple pass/fail evaluation. By introducing Refusal Drift and Compliance Error, it provides a way to measure how a model's safety fails.

**Weaknesses:**

- The paper makes broad claims about "scaling laws" but bases its conclusions on only three open-source, instruction-tuned models.
- The conclusions are drawn from a very small data sample. The adversarial fine-tuning used a training set of only approximately 500 prompt-completion pairs , and the final evaluation for all models and attacks was performed on a unified held-out set of 100 adversarial prompts. This small benchmark size limits the statistical power of the conclusions.
- The authors acknowledge their new metrics have weaknesses. RRR and CE depend on rule-based detection and LLM-assisted auditing, which are prone to false positives and negatives.
- The paper reads odd even after, as the authors have stated, using GPT-5 to assist writing. For example, the related work section reads like disconnected sentences without clear logic flows. A further round of revision will be helpful to this paper.

**Questions:**

See above.

---

### Official Review · Reviewer_qBEE · 2025-11-05

**Soundness:** 2
**Presentation:** 1
**Contribution:** 1
**Rating:** 2
**Confidence:** 4

**Summary:**

The paper studies the scaling properties of adversarial robustness to prompt-based and finetuning attacks by studying TinyLlama-1.1b, Phi-3-mini-3.8b, and Qwen2-7b, using three different metrics to understand refusals.

**Strengths:**

Originality: The paper studies different metrics that capture more qualitative changes in how the model's refusal mechanisms break down.
Clarity: The paper is reasonably understandable for what it covers.

**Weaknesses:**

W1 (Quality): The paper uses one example from each model family to achieve its scaling, rather than picking one model family (such a Qwen3) and varying the size within that model family. This makes it harder to understand what is a result of model scale, as opposed to other changes in the training recipe.
W2 (Significance): The paper is missing related work on scaling trends in adversarial robustness (such as https://arxiv.org/pdf/2407.18213 , or its early citations).
W3) The paper is substantially below the page count, and appears incomplete (i.e. listing prompt-based attacks in the threat model, but not presenting results on them).

**Questions:**

Q1) Are the Figure 1 subcaption headings mislabeled? It seems like (b) should be the caption for the far left graph, and that the middle graph should have a new caption describing refusal drives.
Q2) Are the results for prompting attacks missing? Figure 1 and Table 1 seem to only describe the LoRA finetuning attack.

---

### Note · Program_Chairs · 2026-01-17
**Submission Desk Rejected by Program Chairs**

The following references in this submission do not refer to real documents and/or have major errors in bibliographic information:

 Katherine Hsu et al. Safelora: Safety-preserving parameter efficient fine-tuning. In NeurIPS, 2024.
Basem Elesedy et al. Lora-guard: Detecting jailbroken lora adapters. In ICLR Workshop, 2024.